# Continuous Land Cover Change Detection in a Critically Endangered Shrubland Ecosystem Using Neural Networks

**Glenn R. Moncrieff** [1,2] 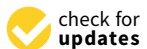

1   Fynbos Node, South African Environmental Observation Network, Private Bag X7, Rhodes Drive, Claremont 7735, South Africa; glenn@saeon.ac.za
2   Centre for Statistics in Ecology, Environment and Conservation, Department of Statistical Sciences, University of Cape Town, Private Bag X3, Cape Town 7701, South Africa

**Abstract:** Existing efforts to continuously monitor land cover change using satellite image time series have mostly focused on forested ecosystems in the tropics and the Northern Hemisphere. The notable difference in spectral reflectance that occurs following deforestation allows land cover change to be detected with relative accuracy. Less progress has been made in detecting change in low productivity or disturbance-prone vegetation such as grasslands and shrublands where natural dynamics can be difficult to distinguish from habitat loss. Renosterveld is a hyperdiverse, critically endangered shrubland ecosystem in South Africa with less than 5–10% of its original extent remaining in small, highly fragmented patches. I demonstrate that classification of satellite image time series using neural networks can accurately detect the transformation of Renosterveld within a few days of its occurrence and that trained models are suitable for operational continuous monitoring. A dataset of precisely dated vegetation change events between 2016 and 2021 was obtained from daily, high resolution Planet Labs satellite data. This dataset was then used to train 1D convolutional neural networks and Transformers to continuously detect land cover change events in time series of vegetation activity from Sentinel 2 satellite data. The best model correctly identified 89% of land cover change events at the pixel-level, achieving a f-score of 0.93, a 79% improvement over the f-score of 0.52 achieved using a method designed for forested ecosystems based on trend analysis. Models have been deployed to operational use and are producing updated detections of habitat loss every 10 days. There is great potential for continuous monitoring of habitat loss in non-forest ecosystems with complex natural dynamics. A key limiting step is the development of accurately dated datasets of land cover change events with which to train machine-learning classifiers.

**Keywords:** land cover change; land cover monitoring; deep learning; Renosterveld; threatened ecosystems; Sentinel 2; planet labs

## 1. Introduction

Land cover change is the single largest cause underlying the global biodiversity crisis [1]. It is estimated that at least 21% of all terrestrial plant extinctions are directly related to agriculture alone [2]. A core component of efforts to halt biodiversity loss and climate change are remote sensing-based monitoring systems built upon satellite-derived earth observation data [3–5]. These systems allow up-to-date reporting on trends in habitat loss, and in some instances, near-real time alerts that may aid interventions and enforcement [6,7]. Typically, temperate and tropical forests are the focal ecosystems for applications, examples of which include Global Forest Watch (accessed on: 28 April 2022) and Terra-i (accessed on: 28 April 2022).

While forests are indeed globally important stores of carbon and home to myriad species, far less attention has been paid to developing remote sensing-based monitoring for use in open ecosystems where trees may be present but not dominant, such as savannas, grasslands, shrublands and deserts. Open ecosystems cover a large proportion of the

globe, make up 40% of the global total ecosystem organic carbon and harbour a substantial proportion of the world's biological diversity [8,9]. Failure to develop capacity for change detection in open ecosystems would exclude the majority of the earth's terrestrial surface from monitoring. Detecting abnormal change in forested ecosystems is relatively simple in comparison to open ecosystems. The spectral signature of a closed-canopy forest stands in stark contrast to the sparsely vegetated or bare ground that remains after deforestation. In contrast, the natural vegetation state in open ecosystems can vary dramatically due to natural disturbances, long-term trends or cyclical functions, such as those relating to fire, seasonality and inter-annual weather variations [10,11]. This variability found within healthy ecosystems can obscure spectral change that occurs when intact natural vegetation is converted to non-natural land use.

The Cape Floristic region (CFR) in South Africa is home to some of the world's most floristically diverse open ecosystems [12,13]. Within the CFR, the most threatened vegetation type is the grassy shrubland known as Renosterveld [14,15]. Renosterveld is restricted to relativity nutrient-rich soils in the south and west of the Western Cape Province. It is noteworthy for its exceptionally high diversity of bulbs and—due to widespread conversion of intact Renosterveld to agriculture—perhaps the highest concentration of plants threatened with extinction in the world [15–18]. Only 5–10% of its original extent remains, with much of this contained in small isolated fragments [19]. Despite legislation prohibiting the destruction of Renosterveld habitat, loss is ongoing [20]. It is estimated that at current rates Renosterveld will become extinct as an ecosystem within the 21st century [15]. The primary cause of this decline is the expansion of intensive agriculture (primarily for the cultivation of grains and oil-seed such as wheat, barely and canola), and overgrazing by livestock.

A remote sensing-based monitoring system, such as is available for many forest ecosystems worldwide, may help to reduce rates of Renosterveld loss by aiding in the identification of landowners responsible for habitat loss. If information is provided with low latency between the onset of ploughing and its detection, there may be potential to intervene while habitat destruction is underway. Continuous change detection approaches attempt to assimilate and analyze new remote sensing data as they become available, and have the potential to provide near-real time alerts on land cover change. A typical pattern used for continuous change detection involves analyzing trends in historical vegetation activity and building a model that can accurately forecast the expected reflectance of intact forests. Predictions are then compared with recently acquired observations [21–24]. When observed vegetation activity falls outside the range expected for natural vegetation, this is taken to be indicative of forest loss. Trend analysis methods such as continuous change detection and classification (CCDC) used to model expected vegetation activity typically assume a relatively predictable progression of vegetation activity in intact ecosystems that can be well described by linear models with harmonic terms to capture seasonality [22]. Some algorithms such as BFAST-monitor can incorporate exogenous regressors such as precipitation, allowing the expected ecosystem state to vary depending on environmental conditions [21]. However, these still assume linear dependencies between predictor and response variables. The resulting land cover change detections, while impressive for stable ecosystems such as forest, are prone to commission errors in dynamic ecosystems where natural variability can easily be mistaken for land cover change and omission errors when disturbance is subtle. These problems are particularly prevalent in non-forest ecosystems with natural dynamics such as fire, aridity and herbivory [10].

When ground-truth data that distinguish areas where habitat loss has occurred are available, land cover change can be directly classified from time series of vegetation activity using supervised learning. Direct classification using labelled and dated events removes the need to specify a model for expected vegetation activity, as the spectral signature of the change events can be learned from the data [25,26]. For example, Hansen et al. [27] trained decision-tree classifiers to detect forest disturbance in the tropics, comparing recent imagery to a stable reference period. Their system is widely used for forest monitoring

in the tropics and has been incredibly impactful. These approaches are, however, limited to producing updates on land cover change at discrete intervals (often annually), as the training data typically do not specify the exact date on which change occurred, but rather that change occurred at some unknown point between two fixed points in time (e.g., January and December). While this provides useful information for reporting on habitat loss, this information is usually only obtained months or years after the reported change has occurred [26,28].

If a precise date for a land cover change event is available, supervised methods can be trained to classify change using input data covering a period of time shorter than year ( Figure 1). Furthermore, if these labelled events occur at a variety of points in time throughout the year (and potentially across multiple years), models that can recognise the signal of land cover change regardless of the time of year and are robust to intra- and inter-annual natural variability can be trained [29]. These models can then be applied continuously as new data become available and produce near-real-time land cover change alerts. They would also be less prone to errors than change detection methods requiring the ecosystem to follow a predictable pattern (e.g., Verbesselt et al. [21], Zhu and Woodcock [22], Zhu et al. [24]), as the models could theoretically learn to distinguish the spectral patterns indicative of land cover change from natural disturbances at any point in time if shown sufficient examples.

Machine-learning approaches such as random forests and support vector machines have been widely used to directly classify land cover change using multi-temporal remotely sensed input data [26,30–32]. These methods do not, however, explicitly take into account the temporal structure within the data. Deep neural networks have the potential to model more complex patterns than tree- or kernel-based methods. Recent advances that have reduced data requirements and training time of neural networks have seen large increases in their performance and more widespread adoption. They now exceed state-of-the-art performance in a wide range of domains ranging from image classification and segmentation to natural language processing [33]. Neural networks specifically designed for classification of sequential data such as convolutional neural networks (CNN) or Transformers have been shown to improve performance on time series classification tasks relative to other machine-learning methods [34]. They have recently been being applied to land cover classification based on time series of remote sensing imagery with great success [35–37] and direct classification of land cover change from paired images [38]. They now routinely outperform other approaches on a range of remote sensing related tasks [39,40].

Here I describe the design and performance of a system for continuous land cover change detection in a critically endangered shrubland ecosystem, the Renosterveld of South Africa. Using a dataset of precisely dated land cover change events covering multiple years, two different neural network architectures are trained and evaluated against tree-based and trend analysis methods commonly used for land cover change detection and monitoring in forested ecosystems. Model building and design choices are motivated by an intention to produce a system that can be deployed for operational use by conservation management and law enforcement agencies. This demands a system that is robust to the natural variability within this ecosystem, can assimilate and make predictions on new data as it becomes available, and is able to detect habitat loss with minimal delay.

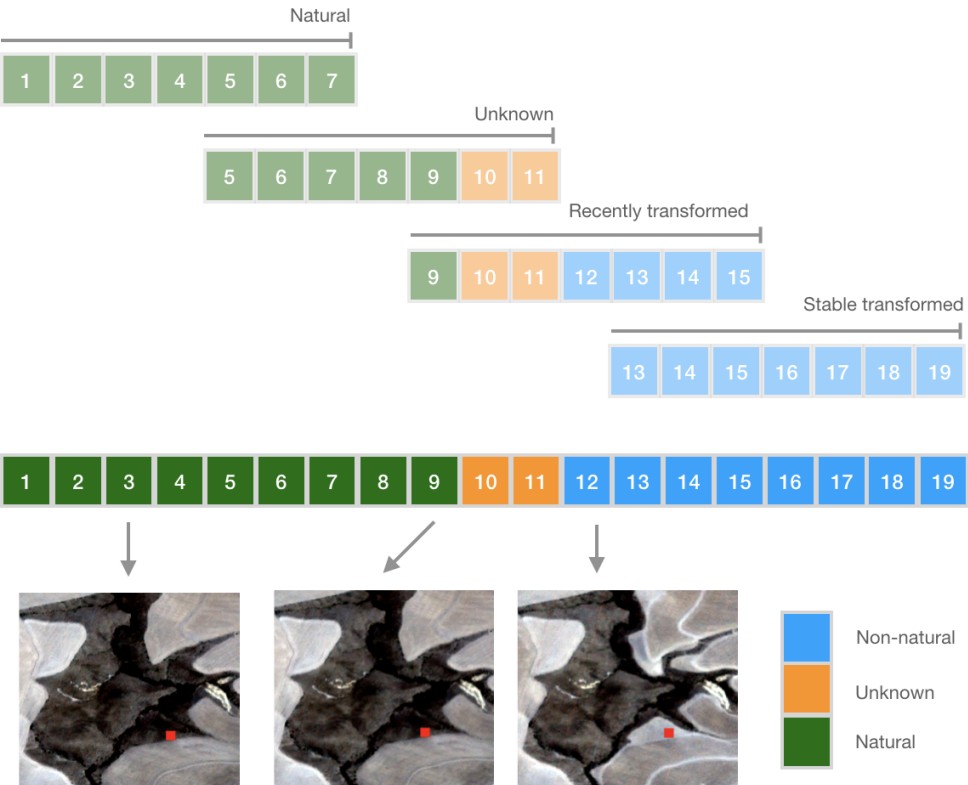

**Figure 1.** The approach used for labelling time series' to enable continuous classification of change. Each step in the sequence of 19 observations represents an observation obtained for a single pixel at a regular interval (e.g., 10 days). The land cover label for the pixel (identified by the red point in the images) can either be natural, unknown or non-natural. This label is obtained through inspection of time series of Planet Labs Planetscope satellite imagery. For the example pixel indicated here, an image obtained at step 3 indicates that the land cover is natural on this date. The image at step 9 indicates the last date on which natural land cover is observed. The first time at which non-natural land cover is observed is at step 12. Hence, it is not known whether the land cover was natural or non-natural at step 10 and 11. Using a window length of 7 observations and forward step of 4, 4 labelled sequences can be extracted from the full time series. Each sequence is assigned the label of the most recent observation and used to train and evaluate machine learning classifiers. These models can then be applied continuously as new data become available to produce near-real-time land cover change alerts.

## 2. Materials and Methods

### 2.1. Study Region

The study region is limited to the extent of the Renosterveld bioregion in the Overberg district municipality in the Western Cape Province of South Africa (Figure 2) [41]. The choice of region was determined by the critically endangered status of Renosterveld vegetation, the availability of manually digitized land cover data for the Overberg municipality and partner conservation organizations in the region. The Overberg municipality covers an area of of 12,241 km$^2$, with Renosterveld covering 4711 km$^2$ within this region naturally. Detailed mapping of remaining natural vegetation in the Overberg based on 2003 data identified 656 km$^2$ of the original extent of Renosterveld vegetation in the Overberg remaining.

### 2.2. Land Cover Change Events

Using random forests for change detection on Sentinel 2 imagery and subsequent detailed verification with very high resolution imagery, Moncrieff [20] identified all conver-

sion of Renosterveld to non-natural land cover between January 2016 and January 2020 in the study region. Land cover change was identified within 268 parcels (a spatially contiguous land cover change event) totalling 478.6 ha. Using daily Planet Labs Planetscope data, dates were assigned to each event indicating the closest possible timing for the occurrence of the land cover change event. It is not possible to assign every event to a single day, as often multiple days or even weeks elapse between the beginning and end of a land cover change event. Furthermore, clouds and haze can reduce the frequency at which clear observations are available. Thus for each land cover change event, two dates were assigned—the latest date on which the presence of natural vegetation could be confirmed, and the earliest date on which the absence of natural vegetation could be confirmed (Figure 1). These two dates occur within one week for 21% of all events, and within two months for 87%. Here a sample of these data were used where land cover changes dates were assigned with the highest confidence, leaving 185 events and 219 ha of land cover change.

This dataset contains only events in which complete conversion of intact natural vegetation occurred and where transformation into non-natural land cover could be dated within a short period of time. It thus excludes gradual degradation occurring over multiple months or years. Fire is a natural process in Renosterveld and is not considered a land cover change event unless it precedes further modification and subsequent conversion to croplands. An additional four events covering 7 ha were identified within the same region between January 2020 and January 2021. These events were used for testing model performance outside of the temporal range of training data, providing a more accurate assessment of operational performance. To augment data with additional observations of stable natural vegetation, parcels of vegetation that remained natural over the entire study period were added. This added an additional 2590 ha dataset covering the period 2016–2019 and 24 ha to the testing dataset covering 2020. These no-change parcels were selected from the Von Hase et al. [19] map of remaining Renosterveld, buffered to ensure that they did not occur within 100 m of non-natural vegetation and manually verified.

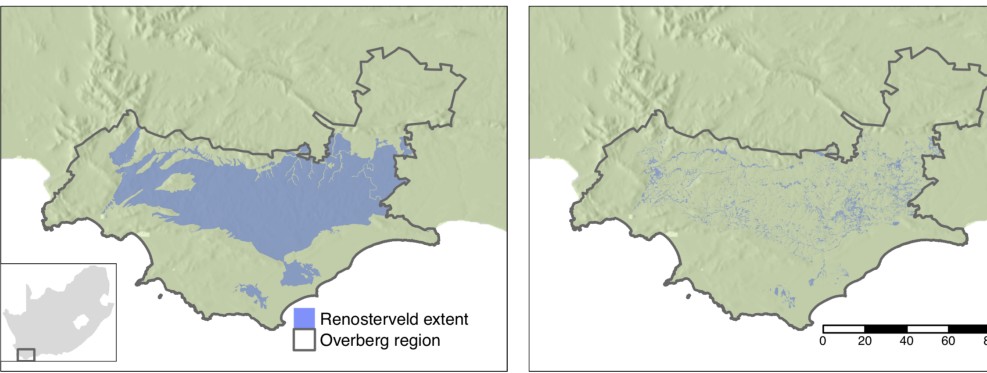

**Figure 2.** The natural extent of lowlands Renosterveld in study region (**left panel**) and the remaining natural and degraded fragments (**right panel**).

### 2.3. Sentinel 2 Time Series

Input time series for classification were created using Sentinel 2 L1C data (S2). For training and validation, all available S2 data up until 31 December 2019 were extracted and preprocessed, whereas testing data covered the period 1 January 2020 to 31 December 2020. S2 images were prescreened and discarded if total scene cloud cover exceeded 50%. Pixel-level cloud masking was performed using the s2cloudless machine-learning-based cloud masks. Pixels with a cloud probability greater than 40% were masked, with this threshold chosen based on visual inspection of the results obtained when using a variety of thresholds. Cloud shadows were also masked and were calculated from cloud masks using cloud and solar geometry combined with dark pixel detection. All S2 bands were retained and the following additional vegetation indices calculated:

Normalized Difference Vegetation index

$$NDVI = (NIR - Red)/(NIR + Red)$$

Enhanced Vegetation index

$$EVI = 2.5 * ((NIR - Red)/(NIR + 6 * Red - 7.5 * Blue + 1))$$

Normalized Difference Water index

$$NDWI = (NIR - SWIR1)/(NIR + SWIR1)$$

Normalized Burn Ratio

$$NBR = (NIR - SWIR2)/(NIR + SWIR2)$$

Normalized Difference Red-edge index

$$NDRE = (NIR - RedEdge1)/(NIR + RedEdge1)$$

Data were exported at 10m resolution with 20 m and 60 m bands resampled using bilinear interpolation.

S2 data were available every 10 days in this region until mid-2017 when data became available at least every 5 days subsequent to the launch of Sentinel 2B. To create regular time series from the S2 image stacks, pixel-level time series were created with a single observation every 10 days. Where multiple images were available within a 10-day period, data from the image with the highest pixel-level NDVI value were used. Where no data was available, values from the previous valid image was used unaltered. In this application, interpolation between observations by averaging is undesirable as it may obfuscate sudden changes in reflectance diagnostic of land cover change. The full record of S2 data for the training and evaluation period over the region covers more than 4 years of data (over 146 observations when using a 10-day interval). For each pixel the full record is split into windows of 180 days (18 observations), with the window shifted forward by 60 days before another time series is extracted (Figure 1). This results in up to 24 separate time series extracted from each pixel. The choice of 180 days was arrived upon through experimentation with both longer and short time series. While it is theoretically possible to detect Renosterveld loss with as few as 2 sequential observations if the change occurs rapidly, few of the labelled events have such precision in determining the actual data on which habitat loss occurred. Longer time series contain more information on the spectral properties of vegetation before and after change, but no additional performance was gained when experimenting with time series longer than 180 days.

### 2.4. Data Labelling

Classification was performed at the pixel-level. The S2 time series for each pixel within a parcel is assigned the label 'natural' if the last date of confirmed natural vegetation fell after the end date of the time series under consideration (Figure 1), or if no change was ever reported within that parcel. If both the last date of confirmed natural vegetation and the first date of confirmed land cover change occur within the 180-day time series, the land cover change event must have occurred within the time series under consideration and the 'recently transformed' label is assigned. If the last date of confirmed natural vegetation occurs within the time series and the first date of confirmed land cover change falls after the end of the time series, it is unclear whether land cover change has indeed taken place, and the data point is discarded. Similarly, if the first date of confirmed land cover change occurs before the start of the time series, land cover change has already taken place. Here a stable non-natural land cover class is not included; thus, these data are discarded.

The resulting labelled dataset was highly imbalanced, with the vast majority (92%) of sampled pixels in the 'natural' class. There was also very high spatial auto-correlation among neighbouring pixels within the same parcel. In addition, successive time series for the same pixel obtained 10 days apart will share 94% of observations. Class imbalance can be addressed using either undersampling of over-represented classes, oversampling of under-represented classes and/or algorithmic approaches that place more value on some observations [42,43]. Here, both undersampling of over-represented classes and an algorithmic approach through the choice of an appropriate loss function was used. To ameliorate spatial autocorrelation, only a subset of pixels within each parcel was used, but only 5% of pixels within parcels where no change occurred were sampled, and 33% of pixels from parcels where change did occur were used. The forward shift of 60 days between extraction of 180-day time series windows reduces temporal oversampling within a pixel, with successive time series sharing only 66% of observations. Subsequent to this data thinning, pixels were stratified by parcel and assigned to training and validation sets using a 70/30 split—that is, 70% of parcels were allocated to the training set and 30% to the validation set. Test data was intended to evaluate model performance in an operational setting and hence did not undergo temporal or spatial thinning, with a new prediction made every 10 days.

### 2.5. Models

Two neural network architectures that have shown excellent performance in classification of land cover from satellite image time series were trained and evaluated along with random forests, a tree-based method commonly used for land cover classification. When training random forests, each time step of each input band was input as a separate feature, resulting in 324 features (18 × 18).

### 2.5.1. TempCNN

Convolutional neural networks learn filters that, when applied to multidimensional input, extract features relevant to a particular downstream task. They have typically been applied to tasks where images are the primary input, and the input shape is 3D (width × height × depth). However, they can also be used on time series data with 2D input (length × depth). TempCNN is a convolutional neural network for land cover classification from satellite image time series where convolutional kernels that extract relevant features in local temporal regions across multivariate time series of individual pixels are learned. Convolutional filters are applied to individual pixels separately, thus input time series are of shape 18 × 18 (times steps × input channels). Convolutional layers are followed by nonlinear activation and batch normalization layers and stacked. Class probabilities are produced by passing the features extracted by convolutional layers to a final dense layer and using the softmax function to rescale outputs to between 0 and 1. TempCNN performs well on a range of land cover classifications tasks and has been adopted by multiple packages for classification of satellite image time series [39,44].

### 2.5.2. Transformer

Transformer models were originally proposed as sequence-to-sequence encoder-decoder models based on the self-attention operation and applied to language translation [45]. Attention mechanisms operate by assigning an importance score (attention) to each element in an input sequence representing its relevance to a particular output sequence [46]. In self-attention, each step in a sequence (for example a word in a sentence, or points in a time series) attends to every other step in the same sequence. Through this process, the relationship between steps in a sequence and complex dependencies within sequences are learned. Self-attention encoders convert sequences to a hidden representation that captures these dependencies. In the traditional application of Transformers to language tasks, self-attention encoders are followed by self-attention decoders. For time series classification only the encoder network is needed. In this work, a Transformer model

that has been evaluated in Rußwurm and Körner [36] is used, where individual pixels' time series' are encoded using stacked Transfomer blocks. A Transformer block transforms an input into a higher level multi-dimensional feature representation by passing inputs through a self-attention mechanism followed by multiple dense layers. Prior to passing input sequences to Transformer blocks, positional encoding is added such that the temporal relationship between observations within a sequence is passed to the self-attention layers. The dimensionality of the output from the final Transformer block is reduced using global maximum pooling, passed to a dense block and finally rescaled to class probabilities using the softmax function. This architecture outperforms many other neural networks and tree-based methods for crop-type mapping using satellite image time series [39].

### 2.6. Model Fitting

Hyperparameter optimization for random forests, TempCNN and the Transformer network was performed using a grid search over 24, 36 and 32 parameter combinations respectively. The best fitting model of each class was then evaluated against test data. For the Transformer and TempCNN, models were trained using mini-batch stochastic gradient descent with the Adam optimizer. Focal loss was used as the loss function as this further serves to ameliorate the impact of class imbalance, down-weighting easy examples and placing more value on difficult examples. TempCNN and Transfomers were implemented in Tensorflow 2 using the eo-flow python package. Random forests were implemented using scikit-learn. Google Earth Engine was used for preprocessing input data [47].

To aid model interpretation, the most important temporal regions identified by the TempCNN model for detecting land cover change were calculated using Grad-CAM++ [48]. Originally developed for use with CNNs and image classification, Grad-CAM++ can easily be used when CNNs are applied to time series classification. Grad-CAM++ calculates a saliency score (importance) for each time step in a time series toward the final prediction made by the model. This is calculated using a weighted average of all positive activation gradients across channels in the final convolutional layer.

### 2.7. Trend Analysis Comparison

The accurately dated land cover change data used here for direct classification through supervised methods is seldom available. Trend analysis approaches to land cover change detection are more generally applicable and widely used [6,49]. A method based on trend analysis is used to evaluate the improvement in accuracy achieved when using direct classification on these data. The approach followed is very similar to the approach outlined by Zhu and Woodcock [22], termed continuous change detection and classification (CCDC). CCDC is widely used for continuous change detection and implemented in platforms such as Google Earth Engine. More recent methods based on similar principles that would improve on this baseline are available [23,50,51], but they are not commonly used on S2 data.

The CCDC-like approach operates by fitting a simple linear model with a single harmonic term to capture seasonal variation using robust regression through iteratively reweighted least squares. The model was fitted to historical NDVI data for every pixel in region known to still contain Renosterveld. The expected NDVI over the test period (1 January 2020–31 December 2020) was then calculated for each day of the year at each pixel. This was compared to the observed NDVI and anomalies calculated and expressed in terms of the RMSE of the fitted regression. If 3 successive observations differ from the prediction by 3 times the RMSE, land cover change is assumed to have occurred. The RMSE thresholds used to trigger detection can be changed to account for ecosystems with greater inherent variability [22]. Different thresholds were experimented with, with 3 times the RMSE providing the best performance here. More sophisticated implementations of this workflow make use of multiple bands, or use parameters learned from labelled land cover change events [51].

## 3. Results

Time series of selected pixels extracted for the entire period covered by the training and validation show the highly dynamic and variable nature of vegetation in this ecosystem (Figure 3). These time series show the fall in minimum NDVI that occurs when Renosterveld is converted to cropland (a-iv) and increase in NDVI seasonality—with NDVI exceeding 0.5 in winter but falling to around 0.1 in summer (a-ii, a-iii). However, at some sites in some years, NDVI fails to reach expected highs in croplands (a-i). NDVI can also be higher in natural vegetation than the winter peak in croplands in some years (a-iv). The higher natural variability in NDVI seasonality in natural vegetation is evident in b-i and b-v, with some sites having little to no NDVI seasonality (b-ii). Average NDVI in sites with natural vegetation can vary from 0.5 to 0.2, and large drops in NDVI unrelated to vegetation change can also occur (b-i). Despite this high variability, a distinctive signal attributable to land cover change is visible, and models are able to use this signal to reliably detect land cover change. The most distinct signal that appears following land cover change is the occurrence of very low NDVI values (e.g., a-vi). These values are lower than the minimum NDVI seen at the mid-summer minimum in natural vegetation and are likely the result of the exposure of bare soil subsequent to vegetation removal.

The best TempCNN model used a dropout rate of 0.8, L2 kernel regularization with a regularization factor of $1 \times 10^{-8}$, He normal kernel initialization and six stacked convolutional layers each with 32 filters and 256 neurons in the final fully connected layer. The best performing Transformer used a dropout rate of 0.5, four layers in the encoder with four attention heads, a model depth/query size of 128, no layer normalization and 256 neurons in the encoder feed-forward layer. The best performing random forest model used 100 estimators, and required a minimum of four samples to split an internal node.

The TempCNN achieved the highest performance, with a recall of 0.89 and precision of 0.96 when identifying recently transformed Renosterveld and an overall f-score of 0.93 (Table 1). The Transformer was only slightly worse than the TempCNN with a recall of 0.88 and precision of 0.97 on the recently transformed class and overall f-score of 0.92, while the random forest was notably worse than both TempCNN and the Transformer, with recall and precision of 0.79 and 0.98 on recent transformation, and an overall f-score of 0.88. The reduction of missed Renosterveld loss events when using the TempCNN over random forests by almost 50% (recall of 0.79 vs. 0.89 implies 21% vs. 11% missed events) is particularly significant for operational performance. All models using direct classification outperformed the trend analysis method, which was far more likely to produce both false negatives and false positives with recall of 0.49 and precision of 0.54 on the recently transformed class and an overall f-score of 0.52.

Performance for random forests, TempCNN and the Transformer declines when their ability to detect recent transformation within 20 days of change is measured. Again the TempCNN performed best, with recall of 0.64, followed by the Transformer with 0.60 and the random forest with 0.50. Trend analysis classification is not possible within 20 days of change, as three anomalous data points are required before change is flagged. This equates to 30 days with the 10-day interval used here.

Figure 4 provides a visual depiction of the time series features characteristic of Renosterveld loss events, determined using Grad-CAM++ on the trained TempCNN. For a single Renosterveld loss event in a pixel viewed from multiple points in time after the event, the same temporal features are focused upon. Across multiple different land cover change events, the network has learned to consistently focus on time periods in which unexpected lows in vegetation greenness occur. These periods occur directly after precipitous declines in NDVI associated with change events. Other more nuanced changes in spectral reflectance may also be contributing to the importance of these temporal regions, as the TempCNN is using all input bands and indices to classify change, though only NDVI is depicted here.

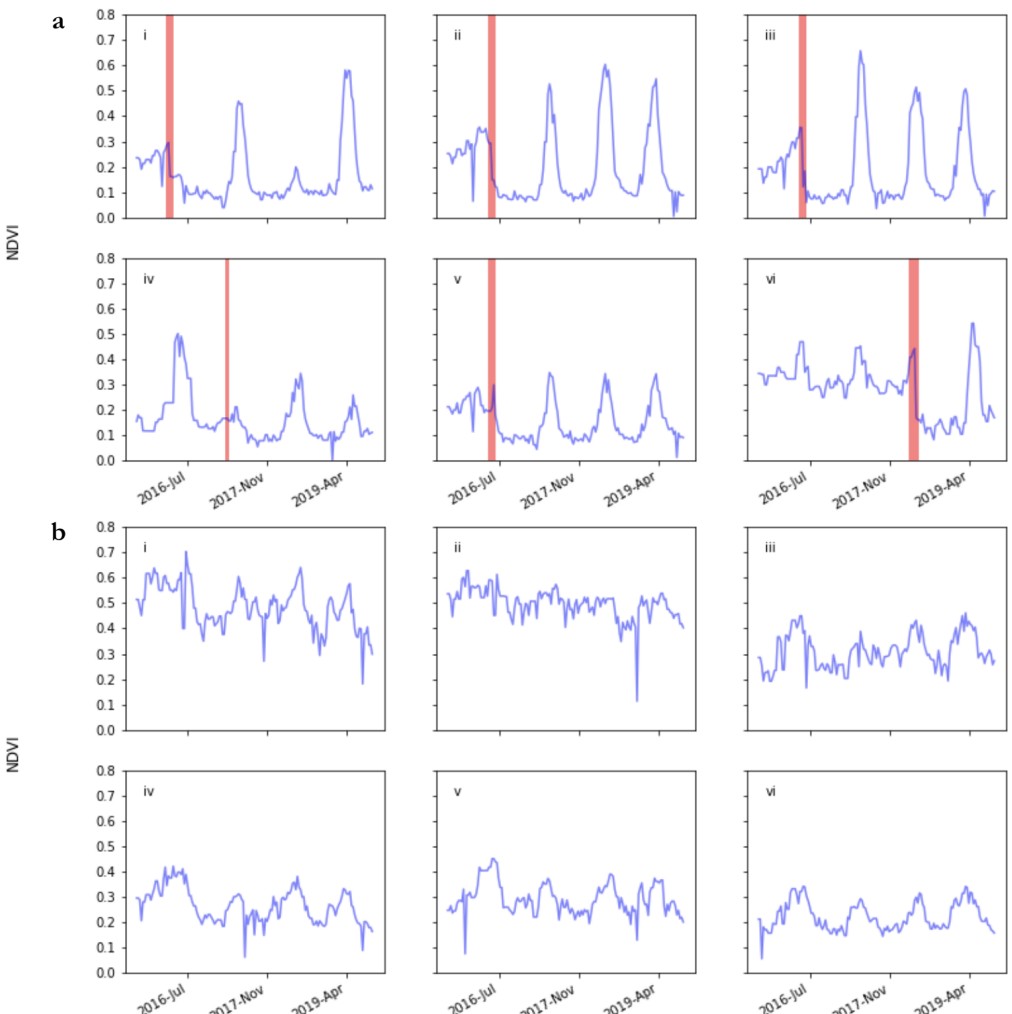

**Figure 3.** Observed NDVI time series for the full period covered by training and validation data for randomly selected pixels from across the study region. Vegetation in all time series begins as natural Renosterveld. In panel (**a**), land cover change events occur in the period highlighted in red, and thereafter the land cover is non-natural. These time series show the fall in minimum NDVI that occurs when Renosterveld is converted to cropland (iv) and increase in NDVI seasonality—with NDVI exceeding 0.5 in winter but falling to around 0.1 in summer (ii, iii). However, at some sites in some years, NDVI fails to reach expected highs in croplands (i). NDVI can also be higher in natural vegetation than the winter peak in croplands in some years (iv). In (**b**), vegetation remains natural for the entire time series. The higher natural variability in NDVI seasonality is evident in (i) and (v), with some sites having little to no NDVI seasonality (ii). Average NDVI in sites with natural vegetation can vary from 0.5 to 0.2, and large drops in NDVI unrelated to vegetation change can also occur (i). These examples serve to illustrate the complex natural variability within natural vegetation and the challenge of extracting reliable identifiers of land cover change from satellite image time series.

A prototype application performing up-to-date inference over the entire study region every 10 days using TempCNN is available at https://glennwithtwons.users.earthengine.app/view/global-renosterveld-watch (accessed on: 28 April 2022), and an example sequence of predictions over 2 months in an area through the course of a land cover change event is depicted in Figure 5. Inference over the entire remaining distribution of Renosterveld in the Overberg region for a single date reveals multiple as-yet unreported areas where Renosterveld has been transformed to cropland within the 6-month time series of satellite data under consideration. Because models are trained to only make predictions in regions where Renosterveld is presently intact, predictions made on other land cover types, or where Renosterveld has already been converted to agriculture, are not reliable. This is not an issue when up-to-date land cover maps are available. Multiple observed cases of incorrect predictions of Renosterveld loss where land cover has in reality remained as agriculture throughout the time series under consideration highlight the need for updating existing regional land cover maps.

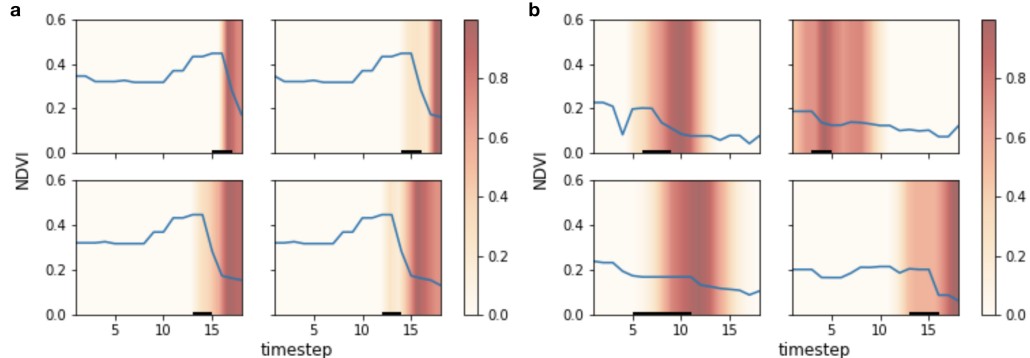

**Figure 4.** Saliency of temporal regions in time series for pixels in which land cover change is predicted to have occurred, determined using Grad-CAM++. Higher saliency scores indicate that the region is more influential in determining that land cover change has occurred. Blue lines show NDVI, though saliency is calculated using all input features. Black underlines show the period over which the actual land cover change event occurred. In (**a**) a single pixel is shown at four points in time, with each subsequent panel shifted forward one step (10 days). In (**b**) time series from four spatially and temporally independent pixels are shown.

**Table 1.** Model performance evaluated on test data. Recall-1-short reports model recall when change has occurred within the first 2 time steps (20 days), of the 18-step (180 days) time series. Performance of the top-performing model for each metric is highlighted in bold.

|  | Accuracy | F1 | Recall-1 | Recall-0 | Precision-1 | Precision-0 | Recall-1-Short |
|---|---|---|---|---|---|---|---|
| Random Forest | 0.98 | 0.88 | 0.79 | **0.99** | **0.98** | 0.98 | 0.50 |
| TempCNN | **0.99** | **0.93** | **0.89** | **0.99** | 0.96 | **0.99** | **0.64** |
| Transformer | **0.99** | 0.92 | 0.88 | **0.99** | 0.97 | **0.99** | 0.60 |
| CCDC | 0.90 | 0.52 | 0.49 | 0.95 | 0.54 | 0.94 | - |

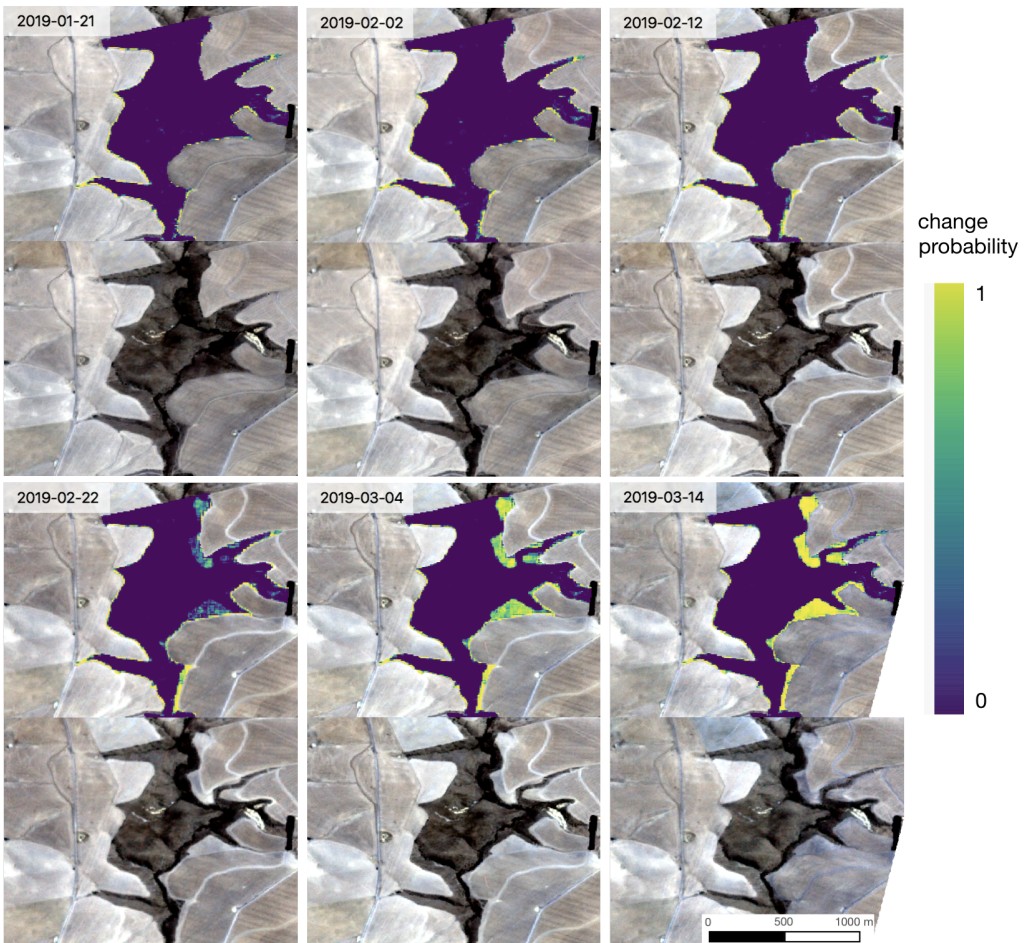

**Figure 5.** Example sequence of predictions through the course of a land cover change event in Renosterveld shrublands of the Overberg region, South Africa centered at 34.33°S, 19.68°E . Upper panels show the predicted probability of land cover change, where predictions are only made in pixels known to have intact natural vegetation at the start of the monitoring period. The lower panels show the most recent Planet Labs Planetscope imagery collected prior to, or on the date for which predictions are made. The detection of Renosterveld loss along the eastern edge of the monitored area is apparent. This detection occurred within 10–20 days of the actual land cover change event.

## 4. Discussion

Shrublands are globally important repositories of biodiversity and play an important role in regulating planetary biochemical cycles [8,9]. In many regions, shrublands are being lost at a rate that exceeds other ecosystems such as forests [52]. Despite this, few studies have attempted to develop algorithms designed specifically for operational monitoring of shrubland loss, ostensibly because many of these ecosystems display complex natural dynamics and have low overall productivity. These characteristics make the methods typically applied in operational forest monitoring perform poorly. This study demonstrates that reliable detection of land cover change in shrublands is possible, and that detection is possible within days of land cover change events, thus facilitating the use of these algorithms for operational monitoring and enforcement. The key component of this is the continuous use of direct classification, enabled by the availability of hand-labelled change events with the date of each event precisely determined.

Direct classification of change using accurately dated events to train models is rarely used for continuous detection of land cover change. The labels needed to train models require detailed examination of very-high resolution satellite or aerial imagery and dense time series. Very-high resolution data may not be available for the required region, time

period or frequency, and the costs associated with these will be prohibitive for many potential use cases. Even when all the necessary data is available, the data labeller must have intimate knowledge of the ecosystem in question to determine whether an observed event is indeed land cover change. For example, Renosterveld is a naturally fire-prone ecosystem and burned vegetation is not necessarily indicative of land cover change. These limitations should not, however, preclude the development of labelled datasets. The availability and cost of very-high resolution and high temporal frequency imagery is improving dramatically, and new programmes are broadening access to these data (e.g., the Norway International Climate and Forest Initiative—Planet partnership). Advances in deep-learning performance on land cover classification when labelled data are limited will further reduce data requirements [53,54]. The investment of time required to acquire labels is indeed significant, with between 5 and 15 min required to confidently assign dates to each event in this study [20]. However, for critically endangered ecosystems that are predicted to become extinct within decades based on current trajectories, surely this investment is justifiable. Particularly if there reason to believe that an operational land cover change monitoring system will slow or reverse these trends.

The volume of labels required in the example presented here was surprisingly small—just 219 ha of dated land cover change events. This is sufficient to produce large improvements in performance in deep-learning-based classifiers relative to random forests which typically require less data to train. While it is highly unlikely that a supervised approach will be applicable in any region beyond the domain of the training data (thus requiring further painstaking data collection and labelling), meta-learning and transfer-learning approaches ought to reduce data requirements for application to new disturbance types and new geographies [55]. Given multiple labelled datasets of dated land cover change from different geographies or ecosystems, models can be trained such that they are initialized optimally and learn rapidly when confronted with a new regime. Particularly promising is model-agnostic meta-learning, which does not necessitate custom designed models for this task but rather can make use of widely adopted models such as TempCNN or Transformers [53,56].

The models developed here are not only limited in their spatial domain of applicability, but also in the types of land cover change events which they are capable of detecting. Only a single land cover change can be identified—-the sudden and destructive conversion of natural Renosterveld to non-natural land cover. Other types of change can occur in this landscape, such as gradual degradation through overgrazing or the spread of invasive alien plants. It is possible that the same approach could be used to diagnose these types of changes. This would, however, require a significant increase in the amount of training data and longer input time series, as these types of change occur over periods longer than the 6-month windows input into classifiers. The choice to focus on a single type of rapidly occurring change allowed relatively high performance to be achieved with limited training data and is focused on the actions considered most severe by law enforcement in the region.

A major improvement achieved through the approach presented here is the ability to detect change within a few days of the actual event. Rapid identification is critical in many intended applications of near-real-time continuous land cover change monitoring—such as law enforcement and management intervention. Other approaches typically require multiple consecutive observations confirming the change has occurred before a positive identification is made [22,23]. Generally, any land cover change will be complete by the time these detections are reported and much evidence lost or destroyed. Here change is identified with relative skill with only 2 observations subsequent to the actual change event. In certain applications, this may be sufficient to lead to an on-the-ground intervention that could prevent further damage. Accuracy improves as further observations are acquired; hence, it is possible to delay the identification of change depending on the cost/benefit of any particular application.

The improvement in performance obtained when using deep-learning-based time-series classifiers vs. random forests is likely a result of their ability to retain temporal

information and extract features based thereon. Previous work comparing the performance of Transformers and TempCNN for land cover classification found Transformers to generally perform better than TempCNN, particularly when minimal prepossessing and no atmospheric correction is performed [36,39]. Here, extensive feature engineering has been conducted and cloud masking performed to remove noisy data, potentially resulting in the convergence in performance between TempCNN and the Transformer. A range of machine-learning methods that are not based on neural networks have been developed for the classification of time series data, and there may be potential performance gains when using these methods, particularly when training data are scarce [57,58]. Gains in performance cannot, however, come at the costs of increases in model inference time to the extent that prediction cannot be made at the required latency or within available budgets. This is particularly relevant because monitoring will often be required over very large regions. Using the TempCNN model fitted here, prediction over the entire remaining range of Renosterveld in the Overberg region can be achieved within 120 CPU hours. The full prediction pipeline for preprocessing, predicting and postprocessing is achieved at minimal cost using tools such as Google Earth Engine and Google Cloud Dataflow.

## 5. Conclusions

The potential of continuous change detection using deep learning on satellite image time series, particularly in non-forest ecosystems, remains largely unexploited. This study demonstrates that when labelled data are available, accurate and precise detection of land cover change is possible at low latency. Obtaining the labelled data does indeed require significant effort, and it is likely that independent datasets and models will be required for different ecosystems. This should not discourage the further development and operationalization of this approach. Both deep learning and the explosion of remote sensing data have been lauded as having great potential to address biodiversity and the climate crises. For these technologies to truly have an impact, they must find application beyond the forested ecosystems of the world and embrace the complexity of open ecosystems.

**Funding:** This research was supported in part by by the National Research Foundation of South Africa through (Grant No. 118593) as part of the RReTool: Rapid and repeatable tools for monitoring and mitigating global change impacts on natural resources project, the Group on Earth Observations-Google Earth Engine Programme, and the NASA Ecological Forecasting Team Applied Sciences Program (80NSSC21K1183).

**Data Availability Statement:** Code and data used in this analysis are available at https://github.com/GMoncrieff/renosterveld-monitor (accessed on: 28 April 2022) Due to ongoing criminal investigations location data has been removed to protect the identity of landowners where necessary.

**Acknowledgments:** I am very grateful for the support and encouragement given by Odette Curtis and her passion for conservation in this region. Marcel Gietzmann-Sanders helped produce the prototype operational monitoring application. Multiple individuals from the Western Cape Department of Environmental Affairs and Development Planning are acknowledged for their input, in particular, those from the Environmental Law Enforcement directorate.

**Conflicts of Interest:** The author declares no conflict of interest. The funders had no role in the design of the study; in the collection, analyses, or interpretation of data; in the writing of the manuscript, or in the decision to publish the results.

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
