# Peer review of "Continuous Land Cover Change Detection in a Critically Endangered Shrubland Ecosystem Using Neural Networks"

_remotesensing, doi:10.3390/rs14122766_

Round 1

Reviewer 1 Report

Continuous land cover change detection in a Shrubland ecosystem was studied. A CNN, a Transformer and the Random Forest classifiers as well as a CCDC based method were compared for this type of land cover change detection. The argument of the importance of land cover change in Shrubland areas and the method comparison in this manuscript are novel and interesting. It is strongly recommended that this paper being published in Remote Sensing after a minor revision. Here are some minor comments.

L18-19, There is great potential for supervised approaches to continuous monitoring of habitat loss in ecosystems with complex natural dynamics.  What does it mean?

L170-171, Normalized DifferenceWater index (NDWI), Normalized Burn index (NBR) and Normalized Difference Red-edge index (NDRE).   These indices are not well known definitions. There should be a formula for each.

L207-209, To reduce temporal oversampling within a pixel, time series windows of 180 days are shifted forward by 60 days for every sample taken. How to do? A detailed description is needed.

L257, The approach followed is very similar to the approached outlined by Zhu and Woodcock.     approaches?

Author Response

We have attended to all the issues highlighted by the reviewer by fixing the minor errors pointed out, elaborating on how spectral reflectance time series were sampled and windowed, and providing full equations for all calculated spectral indices

Reviewer 2 Report

The author presents a method for continuous change detection in a shrubland ecosystem in South Africa. The study is novel both because of the (non-forest) ecosystem type being monitored and because of the detailed reference dataset used. The author compares different machine learning and CCDC approaches which is useful for a broader audience. I have only minor comments which I think could improve the readability and impact of the paper. Otherwise I can recommend this for publication.

L1: Insert “monitor” before land cover.

L3: Maybe specify “spectral” reflectance here.

L8: Not clear to the reader what “direct classification” means here.

L13: “multivariate…” this phrasing is awkward and confusing. Suggest revising.

L70: Here and in other places in the introduction I would recommend that you refer to different methods by their names so that they are recognizable by other practitioners. At the moment the introduction reads very nicely for a broader audience but lacks detail that specialist audience might want. For instance here you should specifically name and refer to Continuous Change Detection and Classification (CCDC). Another algorithm that comes to mind is Landtrendr – would be nice to see how that fits in with the picture here.

L116: Correct “operationally” to “operational”

L156: Was the size of the stable natural vegetation dataset relative to the land cover change dataset in proportion to the estimated intact area vs disturbed area? In other words was your reference dataset balanced? If not, how would you expect this to affect your measures of accuracy?

L166: I am interested to know why 40% was chosen. Can you justify?

L180: Does this mean 90 days before and 90 days after the change event? Please specify.

L197-201: How does this affect the training data sample sizes? Would be good to express this in terms of hectares as in section 2.1 or in terms of pixel numbers.

L205: It would be good to ground this in the literature on class imbalance and different techniques to address it. Please motivate your choice. Maybe refer to https://link.springer.com/article/10.1186/s40537-019-0192-5

L214-236: I know little about deep learning so cannot comment much there. But I am familiar with Random Forest and I am interested to know how you fed your time series data into the RF model? Please elaborate on how you did this.

L276: Unnecessary to have this sentence in the results. This reads like the figure title.

L281: What is the nature of this “distinctive signal”? Is it the precipitous drop in NDVI? In Fig A iv there appears to be no drop. Is there a common time of year that these events occur? What is the model relying on to detect change? I see you go into this in L305-313 and Fig. 4 (very nice by the way!), but I think you can give a bit more detail here.

L294: Was it really “significantly” worse? Seems like very small difference to me. Maybe try frame the difference in terms of operational monitoring. How much accuracy is “significant” for conservation managers for example.

L322: I think this is a very interesting point about the generalizability of your model. How distinct is renosterveld from other fynbos veg types? Did you test (even qualitatively) how well your model performs when making inference in other veg types? Would be interesting to know.

Figure 4. It would be nice to see when the land cover change event actually occurred in these graphs. Like in Fig. 3 A.

L338: I am still not sure what “direct” means here (as in the abstract). Would be good to define this in the introduction.

L355: Could you give a time estimate for how long it took you? Did you use Planet Explorer to do this? Would be useful to know for others wanting to do the same.

Author Response

I am grateful to the reviewer for the detailed feedback. All issues have been attended to. Below are detailed responses to each point

L1: Insert “monitor” before land cover.

Done

L3: Maybe specify “spectral” reflectance here.

Done

L8: Not clear to the reader what “direct classification” means here.

'Direct' has been removed from the abstract, but is properly explained in the introduction now

L13: “multivariate…” this phrasing is awkward and confusing. Suggest revising.

Done

L70: Here and in other places in the introduction I would recommend that you refer to different methods by their names so that they are recognizable by other practitioners. At the moment the introduction reads very nicely for a broader audience but lacks detail that specialist audience might want. For instance here you should specifically name and refer to Continuous Change Detection and Classification (CCDC). Another algorithm that comes to mind is Landtrendr – would be nice to see how that fits in with the picture here.

Done, we have also added reference to BFAST-monitor, a related method (L70-76)

L116: Correct “operationally” to “operational”

Done

L156: Was the size of the stable natural vegetation dataset relative to the land cover change dataset in proportion to the estimated intact area vs disturbed area? In other words was your reference dataset balanced? If not, how would you expect this to affect your measures of accuracy?

Additional info and dealing with imbalance in the dataset is now included as well as statistics comparing areas covered by different classes (L158, L169, L228, L231-235)

L166: I am interested to know why 40% was chosen. Can you justify?

Added justification

L180: Does this mean 90 days before and 90 days after the change event? Please specify.

this section has been expanded to avoid creating confusion L203-211

L197-201: How does this affect the training data sample sizes? Would be good to express this in terms of hectares as in section 2.1 or in terms of pixel numbers.

Additional stats on the area covered by each class has been added (L158, L169, L228)

L205: It would be good to ground this in the literature on class imbalance and different techniques to address it. Please motivate your choice. Maybe refer to https://link.springer.com/article/10.1186/s40537-019-0192-5

Additional info and dealing with imbalance in the dataset is now included as well as statistics comparing areas covered by different classes (L158, L169, L228, L231-235)

L214-236: I know little about deep learning so cannot comment much there. But I am familiar with Random Forest and I am interested to know how you fed your time series data into the RF model? Please elaborate on how you did this.

Details added (L249-251)

L276: Unnecessary to have this sentence in the results. This reads like the figure title.

edited

L281: What is the nature of this “distinctive signal”? Is it the precipitous drop in NDVI? In Fig A iv there appears to be no drop. Is there a common time of year that these events occur? What is the model relying on to detect change? I see you go into this in L305-313 and Fig. 4 (very nice by the way!), but I think you can give a bit more detail here.

more detail added L341-344, L373-375

L294: Was it really “significantly” worse? Seems like very small difference to me. Maybe try frame the difference in terms of operational monitoring. How much accuracy is “significant” for conservation managers for example.

further explanation added

L322: I think this is a very interesting point about the generalizability of your model. How distinct is renosterveld from other fynbos veg types? Did you test (even qualitatively) how well your model performs when making inference in other veg types? Would be interesting to know.

this is an important issue to explore further and the subject of current research. however, the results will be presented in a separate paper

Figure 4. It would be nice to see when the land cover change event actually occurred in these graphs. Like in Fig. 3 A.

added

L338: I am still not sure what “direct” means here (as in the abstract). Would be good to define this in the introduction.

clarification added

L355: Could you give a time estimate for how long it took you? Did you use Planet Explorer to do this? Would be useful to know for others wanting to do the same.

added

Reviewer 3 Report

Dear author, 

the manuscript presented is overall well written and structured. Minor clarifications and typos are listed in the attached document. 

However, the concept of Neuronal Networks for remote sensing applications for land cover/change mapping should be introduced in greater detail in the introduction. Furthermore, also in the methodological part the two models presented should be introduced more in depth. 

Finally, some of the figures need more detailed description (see in attached document). 

Author Response

I am greatful for the helpful comments from the reviewer. All minor issues have been rectified.

In addition both the general background on neural networks has been expanded (L108-122) and the specific background to the models I used (L253-288).

Additional explanation has been given to figures (L331-344), and edits made to figures to improve clarify (fig 4, fig 5). A full map of the predictions made over the study region is however not provided as requested. This is because very little is visible at this scale due to predictions only being made in remnants of natural vegetation, and the highly fragmented and sparse cover of natural vegetation over the study region.

Sections of the methods description have been expanded to clarify issues that were previously unclear (L292-296, L203-209)